# Plasma Lipidomics Profiles Highlight the Associations of the Dual Antioxidant/Pro-oxidant Molecules Sphingomyelin and Phosphatidylcholine with Subclinical Atherosclerosis in Patients with Type 1 Diabetes

**DOI:** 10.3390/antiox12051132

**Published:** 2023-05-20

**Authors:** Lidia Sojo, Elena Santos-González, Lídia Riera, Alex Aguilera, Rebeca Barahona, Paula Pellicer, Maria Buxó, Jordi Mayneris-Perxachs, Mercè Fernandez-Balsells, José-Manuel Fernández-Real

**Affiliations:** 1Department of Diabetes, Endocrinology and Nutrition, Dr. Josep Trueta Hospital, 17007 Girona, Spain; 2Girona Biomedical Research Institute (IDIBGI), 17007 Girona, Spain; 3CIBER Fisiopatología de la Obesidad y Nutrición (CIBEROBN), 28029 Madrid, Spain; 4Department of Medical Sciences, School of Medicine, 17003 Girona, Spain

**Keywords:** plasma lipidomics, subclinical atherosclerosis, overweight, obesity, type 1 diabetes

## Abstract

Here, we report on our study of plasma lipidomics profiles of patients with type 1 diabetes (T1DM) and explore potential associations. One hundred and seven patients with T1DM were consecutively recruited. Ultrasound imaging of peripheral arteries was performed using a high image resolution B-mode ultrasound system. Untargeted lipidomics analysis was performed using UHPLC coupled to qTOF/MS. The associations were evaluated using machine learning algorithms. SM(32:2) and ether lipid species (PC(O-30:1)/PC(P-30:0)) were significantly and positively associated with subclinical atherosclerosis (SA). This association was further confirmed in patients with overweight/obesity (specifically with SM(40:2)). A negative association between SA and lysophosphatidylcholine species was found among lean subjects. Phosphatidylcholines (PC(40:6) and PC(36:6)) and cholesterol esters (ChoE(20:5)) were associated positively with intima-media thickness both in subjects with and without overweight/obesity. In summary, the plasma antioxidant molecules SM and PC differed according to the presence of SA and/or overweight status in patients with T1DM. This is the first study showing the associations in T1DM, and the findings may be useful in the targeting of a personalized approach aimed at preventing cardiovascular disease in these patients.

## 1. Introduction

Atherosclerosis is well known to be the most common cause of cardiovascular disease (CVD) worldwide. In patients with diabetes, it is also the leading cause of morbidity and mortality, being the most prominent contributor to the direct and indirect costs of the disease [1]. Clinical presentation clearly differs between type 1 DM (T1DM) and type 2 DM (T2DM). CVD in T1DM presents at younger age, men and women are affected equally at ages lower than 40 years, and it is more diffuse and concentric [2]. Structural and functional microvascular impairments precede the development of CVD in T2DM patients [3,4]. In T1DM patients, microvascular impairments develop several years after diagnosis, and intensified insulin treatment is associated with improvement in skin microcirculation [5]. In addition, vascular alterations initiated by chronic hyperglycaemia [6] are indirectly exacerbated by overweight and central obesity, both leading to insulin resistance and hypertension that further increase the risk of CVD in individuals with T1DM [7].

Atherosclerosis is a complex process that involves different mechanisms including endothelial dysfunction, neovascularization, vascular proliferation, apoptosis, inflammation, and thrombosis [8]. The major lipids found in human atherosclerotic lesions are cholesterol, glycerophospholipids, and sphingolipids [9]. However, whereas high circulating levels of LDL cholesterol constitute well-established risk factors for atherosclerosis progression and cardiovascular events [10], little is known about the role of the other lipid components. Traditional lipid measures provide a limited view of the complex lipid metabolism. Conversely, lipidomics offers a novel and powerful avenue to obtain a holistic analysis of the lipid profiles associated with atherosclerosis and a better understanding of the complex interaction between atherosclerosis and T1DM [11]. Thus, in the last years there has been an increased number of studies using lipidomics approaches that investigate the role of lipids in atherosclerotic CVD risk [12]. Among them, sphingolipids, and particularly ceramides, have been consistently found to be positively associated with ACS, CHF, or a vulnerable plaque phenotype [12,13]. A recent study performing a ^1^H-NMR-derived lipidomics analysis in T1DM patients without CVD identified sphingomyelin as independently associated with carotid plaque presence, while *w*-6 fatty acids and linoleic acid were associated with a higher plaque burden [14].

Sphingomyelin has been shown to inhibit the peroxidation of unsaturated phospholipids and cholesterol and was proposed as a natural antioxidant [15,16]. However, sphingomyelin has also been found to show pro-inflammatory/pro-oxidant properties (see below in Discussion). On the other hand, phosphatidylcholine also behaves as an antioxidant molecule in atherosclerosis [17], although, again, dual pro/antioxidant actions have also been found, as further developed in the Discussion.

The aim of this study is to describe and compare the plasma lipidomics profiles and the antioxidant molecules of patients with T1DM with and without subclinical atherosclerosis in an attempt to identify potential biomarkers. We also explore the associations according to overweight status.

## 2. Materials and Methods

This is a cross-sectional study performed at a tertiary institution providing care to a population of >2000 patients with T1DM.

### 2.1. Study Subjects

One hundred and seven patients with T1DM attending the outpatient clinic of our hospital were consecutively recruited between 2015 and 2019 to participate in a vascular ultrasound examination during their annual medical visit to screen for chronic complications due to diabetes as well as success of CVD-risk-factors treatment. Inclusion criteria were age above 18 years with at least 10 years of diabetes duration or patients older than 40 years with at least 5 years of diabetes duration. Patients with known CVD (confirmed by clinical assessment and review of patient medical records) were excluded from the study. Patients were stratified by sex and after examination were divided into two groups for metabolomic study depending on the presence or not of subclinical atherosclerosis.

The Local Ethics Committee approved the study protocol, and written informed consent was obtained from each participant (approval Code number DIABICS 2020092).

### 2.2. Clinical Assessment

Clinical data included age, sex, weight, height, body mass index (BMI), and waist circumference, which were measured using standardized methods and introduced into the electronic medical record. Blood pressure (BP) (mean of 2 measurements separated by 5 min) was measured using a blood pressure monitor (DINAMAP V100 Carescape^®^) after 10 min of patients being seated. Hypertension (HTA) was defined as BP ≥ 140/90 mmHg. Patients treated with antihypertensive drugs were considered to have HTA, regardless of blood pressure values that were obtained in the clinical exam. T1DM duration, history of micro- and macrovascular complications, and dyslipidemia were recorded, too, as well as medical treatment (statins, antiplatelet drugs, and antihypertensive agents).

Patients were interviewed about smoking habits. Tobacco exposure was defined as smoking at any time throughout life independent of whether patients smoked or not at the time of the study. Tobacco consumption was measured in pack-years, where one pack per day for one year is considered one pack-year.

Central obesity was defined according to the National Cholesterol Education Program Adult Treatment Panel III (ATP III) [18]. Subclinical atherosclerosis (SA) was defined as the presence of at least one plaque in any of the carotid or femoral segments explored. Atherosclerotic burden was defined as the total number of plaques found in a patient.

### 2.3. Carotid and Femoral Ultrasound Imaging

Ultrasound imaging was performed indistinctly by two trained investigators using a high image resolution B-mode ultrasound system (Philips ClearVue 550) equipped with a 15 Mhz linear array probe. Bilateral extracranial carotid trees (carotid common artery (CCA), carotid bifurcation (CB), internal carotid artery (ICA), and external carotid artery (ECA)) were examined to evaluate the presence of plaques both in near and far walls of all carotid segments. The 20 mm segment of the common femoral artery (CFA) proximal to the bifurcation of the deep femoral artery was explored to evaluate the presence of plaques both in the near and far walls. The 20 mm segment of the superficial femoral artery (SFA) proximal to the bifurcation was also explored.

Plaques were defined as a thickening from the intimal–luminal to the medial–adventitial interfaces of >1.5 mm or focal wall thickenings encroaching into the arterial lumen by at least 50% of the surrounding intima-media thickness (IMT) value [19]. 

If there were no atherosclerotic plaques in the carotid explored segments, IMT was bilaterally measured in common carotid, bifurcation, and internal carotid segments, and the average of the values was considered.

### 2.4. Microvascular Complications

Chronic microvascular complications were defined as the presence of either diabetic retinopathy, nephropathy, or peripheral neuropathy diagnosed using standard methods in routine assistance appointments. The presence of microaneurysms or exudates in routine fundus retinography exams; altered, repeated albumin-to-creatinine ratio in urine; and altered thermalgesic and/or vibration sensation at the toes were used to diagnose microvascular complications of diabetes.

### 2.5. Biochemical Measurements

Fasting blood and urine samples were collected and analyzed locally using standardized assays to measure glucose, glycated hemoglobin (HbA1c), lipid profile (including total cholesterol, high-density lipoprotein cholesterol (c-HLD), low-density lipoprotein cholesterol (LDL), and triglycerides), creatinine, glomerular filtration rate (MDRD4), AST, ALT, GGT, and urine albumin-to-creatinine ratio. 

### 2.6. Lipidomics Analysis Using UHPLC-qTOF/MS

Set of lipid standards (Avanti Polar lipids)*:* LPC(18:0); PC(32:0); SM(36:1); DG(36:0); TG(52:3); ChoE(16:0); MAG(18:0).

Set of labeled lipid internal standards (SPLASH, Avanti Polar lipids): LPC(18:1-d7); PC(33:1-d7); SM(36:2-d9); DG(33:1-d7); TG(48:1-d7); ChoE(18: 1-d7); MAG(18:1-d7).

Instrumentation: UHPLC 1290 Infinity II Series coupled to a qTOF/MS 6550 Series, both Agilent Technologies (Agilent Technologies). Analytical column: Kinetex 2.6 µm EVO-C18, 100 Å, 100 × 2.1 mm (Phenomenex).

Sample preparation: For the extraction of more hydrophobic lipids, a liquid–liquid extraction with chloroform: methanol (2:1) based on the Folch procedure was performed by adding ten volumes of chloroform: methanol (2:1) containing internal standard mixture (Lipidomic SPLASH) to plasma. Then, the samples were mixed and incubated at −20 °C for 30 min. Afterwards, water with NACl (0.8%) was added and the mixture was centrifuged at 15,000 rpm. Lower phase was recovered, evaporated to dryness and reconstituted with methanol:methyl-tert-butyl ether (9:1), and analyzed using UHPLC-qTOF (model 6550 of Agilent Technologies, Santa Clara, CA, USA) in positive electrospray ionization mode.

### 2.7. Statistical Analyses

Plasma metabolomics data were first normalized using a probabilistic quotient normalization. Lipidomics profiles associated with atherosclerosis data were identified using machine learning (ML) methods. In particular, we adopted an all-relevant ML variable selection strategy applying a multiple random forest (RF)-based method as implemented in the Boruta algorithm [20]. It has been recently proposed as one of the two best-performing variable selection methods making use of RF for high-dimensional omics datasets [21]. The Boruta algorithm is a wrapper algorithm that performs feature selection based on the learning performance of the model [20]. It performs variable selection in four steps: (a) randomization, which is based on creating a duplicate copy of the original features randomly permuted across the observations; (b) model building, based on RF with an extended data set to compute the normalized permuted variable importance (VIM) scores; (c) statistical testing, to find relevant features with a VIM higher than the best randomly permutated variable using a Bonferroni-corrected, two-tailed binomial test; and (d) iteration, until the status of all features is decided. We ran the Boruta algorithm with 500 iterations, a confidence level cut-off of 0.005 for the Bonferroni-adjusted *p*-values, 5000 trees to grow the forest (ntree), and a number of features randomly sampled at each split given by the rounded down number of features/3 (the mtry recommended for regression). The dependent variables were the presence of carotid plaques (no/yes) or the intima-media thickness (IMT) in a continuous manner. Models were adjusted for age, sex, BMI, waist, diabetes duration, HbA1c, glomerular filtration rate, hypertension, and tobacco exposure.

## 3. Results

One hundred and seven patients with T1DM fulfilling inclusion and exclusion criteria were recruited. Patient characteristics of recruited subjects are shown in Table 1. Mean age of participants (50.5% women) was 52.8 years (SD 12.9) with a mean of 24.5 years of diabetes duration (11.8). Microvascular complications were present in 54.2% of patients, retinopathy being the most frequent complication (retinopathy 34.6%, polyneuropathy 22.4%, and nephropathy 15%). Regarding CVD risk factors, tobacco exposure was present in 41.1% of the subjects, hypertension in 52.3%, and central obesity in 49.5% (ATP III criteria). Of all the patients, 56.1% were on statin therapy.

The combination of lipidomics and machine learning analyses adjusted by age, sex, BMI, diabetes duration, HbA1c, hypertension, glomerular filtrate, smoking, and statin therapy revealed several lipid classes that contributed significantly to different lipidomics profiles in patients with and without subclinical atherosclerosis, such as sphingomyelin, phosphatidylcholine (PC), lysophosphatidylcholine (LPC), triacylglycerol (TG), and diacylglycerols (DG). 

Lipid species significantly and positively associated with the presence of subclinical atherosclerosis were SM(32:2) and some phosphatidylcholine species (PC(30:1) and PC(30:0)), while PC(38:4) was negatively linked (Figure 1A). When we studied the associations according to overweight/obesity status, we confirmed the positive association of one sphingomyelin species, SM(40:2), in subjects with overweight/obesity while we also found a negative association with a lysophosphatidylcholine species among lean subjects (Figure 1B,C).

To further identify lipid signatures associated with atherosclerosis in T1DM patients, we used applied machine learning to identify lipids predictive of IMT (Figure 2A). We found that the significant lipid species associated positively with IMT were phosphatidylcholines (PC(40:6) and PC(36:6)) and cholesterol esters, ChoE(20:5), in all subjects as a whole, and also within subjects with (PC(33:6)] and without overweight/obesity [PC(34:1), PC(34:2), PC(38:2), PC(38:3), PC(38:4), PC(38:5), ChoE(16:0), ChoE(22:6)). Puzzlingly, PC(35:2) was negatively associated with IMT in lean subjects. Triglyceride species were negatively associated with IMT in the whole cohort (TG(53:4)), and also in subjects with (TG(54:4), TG(53:4), TG(58:9)) and without overweight/obesity (TG(50:3), TG(53:3), TG(51:3)). Other diglyceride species (DG(36:3)) were negatively associated with IMT in all subjects and in subjects with overweight/obesity (Figure 2B,C). 

Microcirculation is known to be altered in patients with diabetes [4]. Therefore, we next performed subgroup analyses to identify lipid profiles associated with SA and IMT in patients with and without microvascular complications. SA was associated with lower plasma levels of LPC(O(18:1)) but higher circulating levels of SM(32:2), PC(35:4), and ChoE(17:1) in patients with microvascular complications (Figure 3A). In patients without microvascular complications, SA was positively associated with LPC (20:1), SM(40:4), LPC(O(18:1)), and PC(O(44:5))/PC(P(44:4)) (Figure 3B). Patients with microvascular complications had several negative associations among TG and DG species andIMT, but it was positively associated with LPC (18:1), PC (36:5), and PC(O(36:6))/PC(P(36:5)) (Figure 3C). Conversely, patients without microvascular complications had negative associations among IMT and several PC species, such as PC(37:2) and PC(38:1), but also had a positive association with PC(36:6) (Figure 3D). Notably, IMT was positively associated with ChoE species containing *w*-3 fatty acids such as EPA (20:5) and DHA (22:6) in both patients with and without microvascular complications.

Finally, we performed correlation analyses to identify the relationship among the identified lipid species and clinical variables. Different associations were observed between some lipid species and clinical or analytical parameters (Figure 4). LPC(20:0) showed a negative association with waist, BMI, HbA1c, triglycerides, and PCR and a positive association with cholesterol levels, mainly HDL-c. SM(32:1) showed the strongest positive association with total cholesterol, c-LDL, and c-HDL. DG(36:3) and TG(53:4) were associated positively with BMI, waist, triglycerides, and high-sensitivity PCR and negatively with c-HDL. Phosphatidylcholine showed different associations according to the species: PC(38:3) was associated positively with BMI, waist, triglycerides, high-sensitivity PCR, and markers of liver injury; PC(40:6) and PC(36:6) associated positively with total cholesterol and c-LDL; PC(40:6) associated positively with urine albumin/creatinine ratio; and PC(35:2) associated positively with c-HDL and negatively with waist.

## 4. Discussion

To our knowledge, this is the first study analyzing the plasma lipidomics profiles of patients with type 1 diabetes mellitus. The main findings concern Sphingomyelin, phosphatidylcholine, lysophospholipids, and different di- and triglyceride lipid species. 

### 4.1. Findings Related to Sphingomyelins (SMs)

SM(32:2) was one of the lipid species most linked to atherosclerosis. Circulating levels of SM(32:2) had a positive association with both subclinical atherosclerosis and cholesterol levels (total cholesterol, c-HDL, and c-LDL), especially in subjects with overweight/obesity. Despite having been proposed as natural antioxidants [15,16], SMs have been implicated in inflammatory and immune responses, vascular homeostasis, insulin signaling, and diabetes [22]. SMs are the most abundant sphingolipid in lipoproteins and constitute about 87% of total plasma sphingolipids. They are associated with VLDL/LDL in 63–75%, and with HDL in 25–35%. They accumulate in atherosclerotic plaques in humans [23]. In fact, LDL particles that are present in atherosclerotic plaques have higher sphingomyelin levels than plasma LDL [24]. At the class level, higher plasma concentrations of SMs have been associated with coronary artery disease (CAD) [25] and subclinical atherosclerosis [26]. In addition, patients with unstable angina or acute myocardial infarction had higher plasma SM levels as well as higher sphingomyelin synthase (SMS) activity, compared to healthy individuals [27]. SMs are synthesised from ceramides by sphingomyelin synthases (SMS). SMS2 is mainly located in the plasma membrane [28], and its expression has been positively associated with the development of atherosclerosis [29], whereas reduction of SMs mediated by SMS2 significantly decreases atherosclerosis in mice [30]. In addition, oxidative stress is responsible for the development of endothelial dysfunction, which plays a crucial role in all stages of atherosclerosis, from lesion initiation to plaque erosion and rupture [31]. Notably, treatment with H_2_O_2_, a reactive oxygen species, led to an overexpression of SMS2 that promoted endothelial cell dysfunction through activation of the Wnt/β-catenin signaling pathway [32]. 

Despite this evidence, only one study has assessed the associations between circulating SM levels and cardiovascular disease at a species level. Specifically, among all SM species, only the plasma levels of SM(38:2) were associated with increased odds of CVD [33]. However, their lipidomics method did not include the measurement of SM32:2, which we have identified for the first time as strongly linked to subclinical atherosclerosis in patients with T1DM. Remarkably, a recent study in n = 28 atherosclerotic patients with “soft” (hypoechoic features) or “hard” (hyperechoic features) carotid plaques, determined using ultrasonography, identified three lipid species discriminating both groups in the LDL fraction, specifically PE(38:6), SM(32:1), and SM(32:2), with all of them increased in the “hard”-plaque-type group [34]. 

We also found that SM(40:2) was strongly associated with SA in patients with overweight/obesity. In line with our results, patients with severe coronary calcification had lower serum levels of SM(40:2) than patients with no coronary calcification [35]. The base backbone structure contains either 18:1 or 18:2. Therefore, the N-linked acyl chain in SM(40:2) most likely comprises the very long-chain fatty acids 22:0 or 22:1. Notably, SMS2 knockout, which has been consistently associated with decreased atherosclerosis, significantly decreased the plasma levels of very long-chain SM species, including SM(d18:1/22:0) and SM(d18:1/24:0), with no changes in medium- and long-chain SM species [36]. Interestingly, addition of very long-chain SM species strongly induced the expression of iNOS and ICAM-1 in macrophages, indicating a role of these species in modulating macrophage activation and inflammation. Therefore, besides well-established traditional risk factors for CVD such as high LDL and low HDL, changes in sphingolipids may contribute to the pathogenesis of cardiovascular disease [37,38]. In fact, recent findings pointed at alterations in sphingolipid metabolism as a contributing factor not only for atherosclerosis [39] but also for microvascular complications in patients with T1DM [40,41,42]. 

### 4.2. Findings Related to Ether Lipids

We also found a strong positive association of ether lipid species with SA, specifically PC(O(30:1))/PC(P(30:0)). In addition, we found several ether lipids positively associated with intima-media thickness in subjects without overweight/obesity. Ether lipids are a unique class of glycerophospholipids that have an alkyl chain in the *sn-1* position through an ether bond. Plasmalogens are the most common form of ether lipids containing a vinyl–ether bond at the *sn-1* position [43]. The relatively low dissociation energy of the vinyl–ether bond makes plasmalogens particularly susceptible to oxidation compared to other diacyl glycerophospholipids or polyunsaturated fatty acids, suggesting a role of plasmalogens as potent cellular antioxidants. Therefore, the positive associations of plasmalogens with SA and intima-media thickness may indicate increased activity of protective mechanisms against oxidative stress. In line with these results, patients with NASH had increased serum levels of several plasmalogens compared to individuals with steatosis [44]. Interestingly, 24 h loading of macrophages with oxLDL, a modified LDL that is taken up by macrophages in atherosclerotic plaques and attracts pro-inflammatory cells, led to a significant increase in plasmalogens [45]. In line with our results, treatment with oxLDL increased the levels of PC(P(30:0) and PC(P(34:1)), among others.

### 4.3. Findings Related to Phosphatidylcholines (PCs)

We also found several circulating phosphatidylcholine (PC) species levels associated with both SA and IMT, which differed according to overweight/obesity status. Phospholipids are required for the formation and stability of lipoproteins. Changes in the PC content of various tissues have been described well for metabolic disorders such as atherosclerosis, insulin resistance, obesity, and liver disease [46]. PCs have both anti- and pro-inflammatory activities with different modifications to the polyunsaturated fatty acids (PUFAs) in the sn-2 position. PUFAs are easily oxidized because of the high degree of unsaturation. Notably, we found that PC species containing highly unsaturated fatty acids such as PC(36:6) and PC(40:6) were strongly positively associated with IMT, while PC species with lower unsaturation were associated with lower IMT. In addition, PC(38:4) had the strongest negative association with SA. In agreement with our results, the serum levels of PC(38:4) were decreased in patients with severe coronary calcification compared to patients with mild coronary calcification [36]. 

### 4.4. Findings Related to Lysophosphatidylcholines (LPCs)

The most abundant lysoglycerophospholipids in human blood are lysophosphatidylcholines (LPCs) [47]. LPCs are obtained through the cleaving of PCs via the action of phospholipase A2 (PLA_2_), or by the transfer of fatty acids to free cholesterol via lecithin-cholesterol acyltransferase (LCAT) [48]. Increased activity of PLA2 has been shown to be associated with atherosclerosis and CHD [49,50]. In addition, LPCs play pro-inflammatory, anti-hemostatic, and cytotoxicity roles and are major components of oxidized low-density lipoprotein (ox-LDL). LPCs are also responsible for monocyte migration and induction of cytokine expression in smooth muscle cells as key events in atherogenesis [48]. Despite this background, LPC species had a negative association with subclinical atherosclerosis (LPC(18:2)) and IMT (LPC(20:2)) in our study, but only in lean subjects. Findings from recent clinical lipidomics studies have also been controversial, showing an inverse relationship of LPCs with cardiovascular disease [48]. Hence, patients with severe coronary calcification had lower levels of several LPCs, including LPC(18:2), compared to patients with no coronary calcification [36]. Notably, LPC(18:2) was independently and inversely associated with death and MACE risk in patients with CAD [51]. In this study, LPC was the lipid class most significantly negatively associated with acute coronary syndrome (ACS) in CAD patients. Among the individual LPC species, LPC(18:2) and LPC(20:2) had a consistent negative association with ACS. Consistent with these results, most LPC species, including LPC(18:2) and LPC(20:2), had a strong negative association with both stable CAD (vs controls) and unstable CAD (vs. stable CAD) [52]. These findings may reflect an increase LPCs’ catabolism and clearance from the circulation of patients with coronary artery disease [53]. It has also been suggested that LPCs may maintain stability of atherosclerosis plaque in patients with CAD [51]. In fact, LPC has shown to decrease cholesterol biosynthesis in liver cells [54] and macrophages [55], thereby decreasing cellular cholesterol accumulation and atherogenesis. In addition, mutations that cause decreased LCAT activity (leading to low levels of LPC) have been linked to accelerated atherogenesis in carotid arteries [56]. In fact, low LCAT activity has been described in patients with type 2 diabetes and metabolic syndrome compared to patients without metabolic syndrome [57]. In our study, LPC was associated negatively to BMI and waist circumference, possibly reflecting altered LCAT activity or LPC catabolism in patients with type 1 diabetes and normal weight status.

### 4.5. Findings Related to Di- and Triglyceride Species

Finally, DG(36:3) and TG(53:4) showed a negative correlation with IMT while associated positively with BMI, waist, triglycerides, and high-sensitivity PCR and negatively with c-HDL. Diacylglycerol and triacylglycerol are produced when excess dietary lipids cannot be adequately buffered within adipose tissue stores. Excess fatty acid (FA) delivery and uptake in peripheral tissues such as the liver and skeletal muscle will lead to excess intracellular FA flux. The excess FA will first be activated to form fatty acyl-CoA and then further be converted into triacylglycerol (TAG) within lipid droplets or give rise to diacylglycerols (DAGs) and ceramides through specific metabolic pathways [58]. They have been associated with lipotoxicity and insulin resistance [58].

The main limitation of the current study is its cross-sectional nature, which limits the inference of causation. The findings should be confirmed in larger longitudinal samples of patients with T1DM. Although we adjusted all machine learning models for confounding variables such as age, sex, BMI, diabetes duration, Hba1c, hypertension, glomerular filtrate, smoking, and statin therapy, we cannot rule out the contribution of other potential confounder variables not included in the models. This is a real-world study developed in assistance services. As a result, patients included were not naïve to medications, such as statins or antihypertensive treatments, prescribed to control different cardiovascular risk factors. This fact limits the interpretation of surrogate outcomes such as total serum cholesterol and fractions and urine albumin/creatinine ratio. 

## 5. Conclusions

In conclusion, patients with type 1 diabetes without prior cardiovascular disease showed a differential plasma lipidomics profile depending on the presence of subclinical atherosclerosis and/or overweight status. Sphingomyelin was positively associated with carotid and femoral plaques while phosphatidylcholine showed different associations along with the different species. Lysophosphatidylcholine showed a negative correlation with the presence of subclinical atherosclerosis only in lean subjects. These findings will be useful in the targeting of a personalized approach aimed at preventing CVD in patients with type 1 diabetes. However, these associations need to be considered preliminary and require further validation in larger cohorts.

## Figures and Tables

**Figure 1 antioxidants-12-01132-f001:**
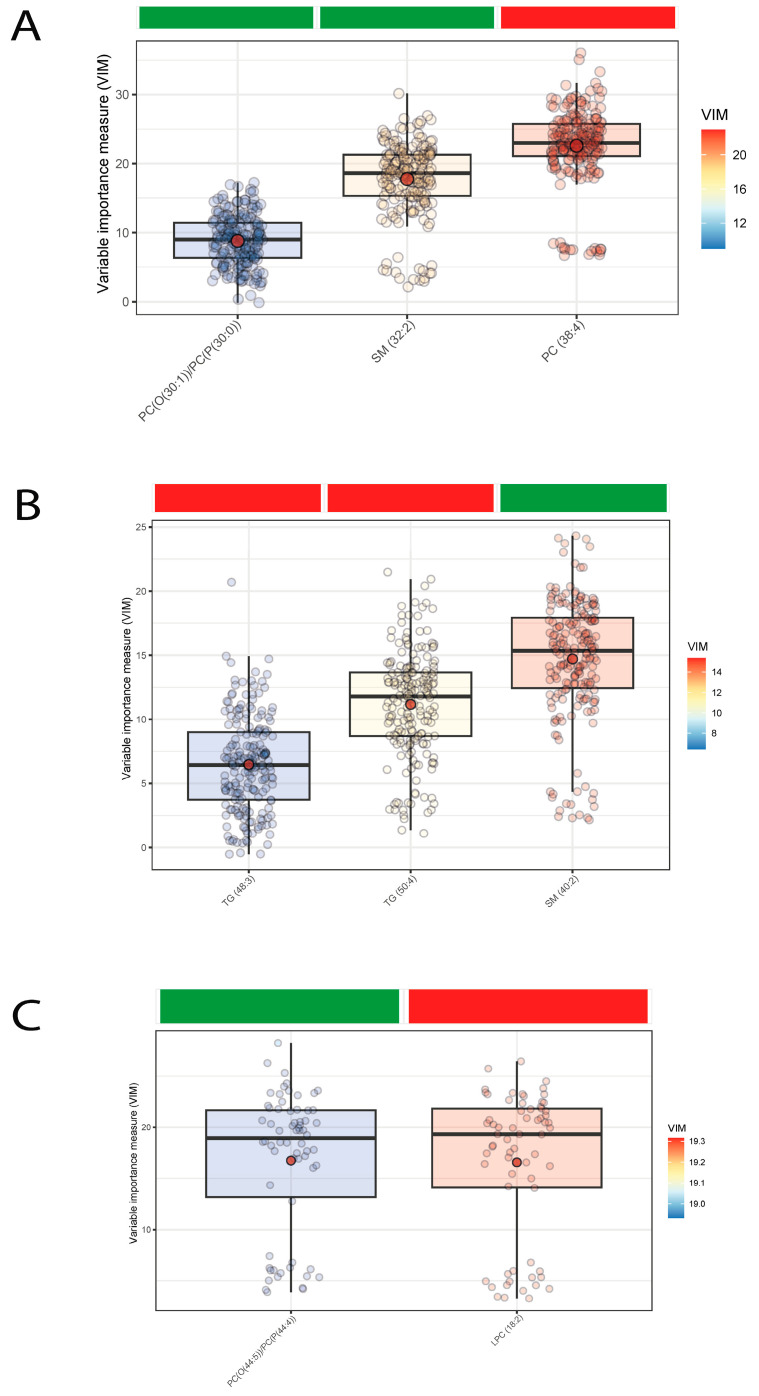
Boxplots of normalized variable importance (VIM) for the lipid species associated with subclinical atherosclerosis. The red dot represents the mean, and the color bar below the plot indicates the sign of the association, with red indicating negative correlation and green positive correlation. (**A**) In all subjects; (**B**) in subjects with overweight/obesity (BMI ≥ 25 kg/m^2^); (**C**) in subjects without overweight/obesity (BMI < 25 kg/m^2^). LPC, lysophosphatidylcholine; PC, phosphatidylcholine; PC(O), alkyl-ether-phosphatidylcholine; PC(P), alkenyl-ether-phosphatidylcholine (plasmalogen); SM, sphingomyelin; TG, triglyceride.

**Figure 2 antioxidants-12-01132-f002:**
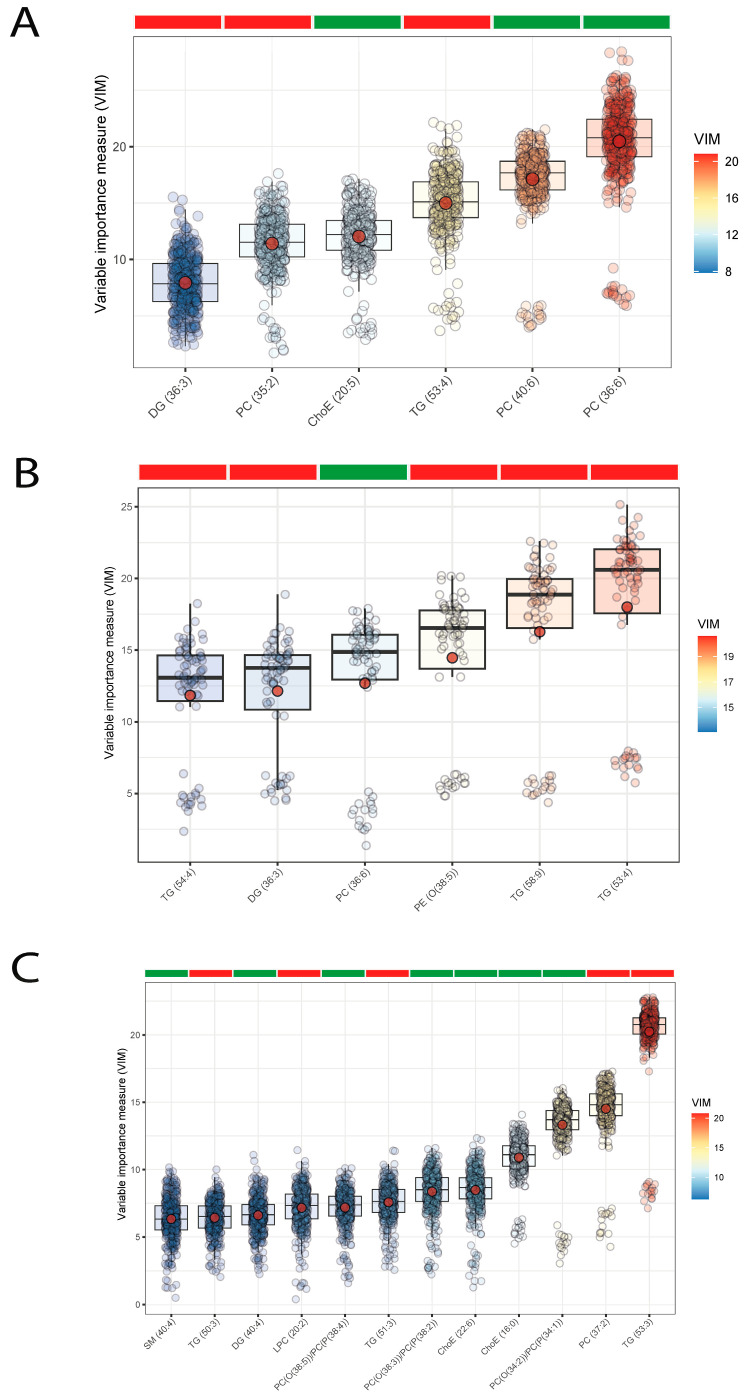
Boxplots of normalized variable importance (VIM) for the lipid species associated with the carotid intima-media thickness (IMT). The red dot represents the mean, and the color bar below the plot indicates the sign of the association, with red indicating negative correlation and green positive correlation. (**A**) In all subjects; (**B**) in subjects with overweight/obesity (BMI ≥ 25 kg/m^2^); (**C**) in subjects without overweight/obesity (BMI < 25 kg/m^2^). ChoE, cholesterol ester; DG, diglyceride; LPC, lysophosphatidylcholine; PC, phosphatidylcholine; PC(O), alkyl-ether-phosphatidylcholine; PC(P), alkenyl-ether-phosphatidylcholine (plasmalogen); PE, phosphatidylethanolamine; SM, sphingomyelin; TG, triglyceride.

**Figure 3 antioxidants-12-01132-f003:**
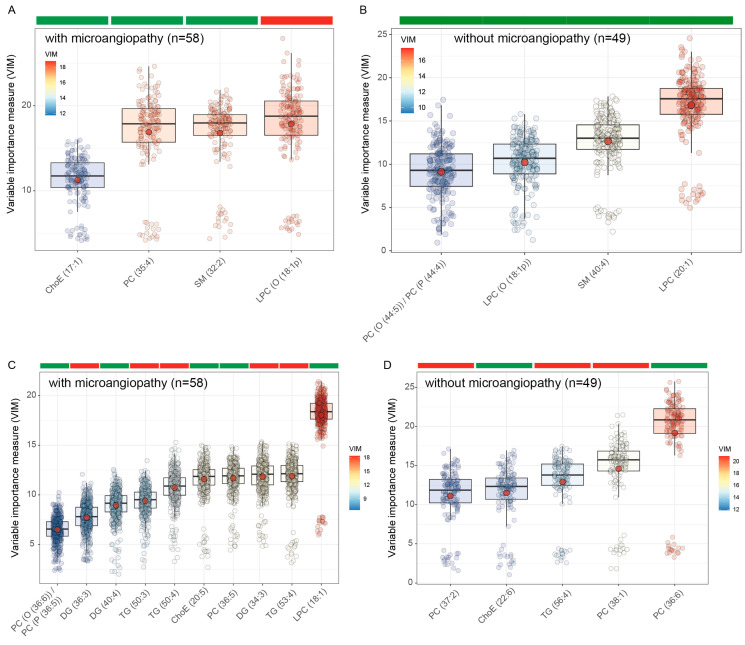
Boxplots of normalized variable importance (VIM) for the lipid species associated with subclinical atherosclerosis (SA) and the carotid intima-media thickness (IMT). The red dot represents the mean, and the color bar below the plot indicates the sign of the association, with red indicating negative correlation and green positive correlation. (**A**) Associations of the lipidome with SA in patients with microvascular complications; (**B**) associations of the lipidome with SA in patients without microvascular complications; (**C**) associations of the lipidome with IMT in patients with microvascular complications; (**D**) associations of the lipidome with IMT in patients without microvascular complications. ChoE, cholesterol ester; DG, diglyceride; LPC, lysophosphatidylcholine; PC, phosphatidylcholine; PC(O), alkyl-ether-phosphatidylcholine; PC(P), alkenyl-ether-phosphatidylcholine (plasmalogen); PE, phosphatidylethanolamine; SM, sphingomyelin; TG, triglyceride.

**Figure 4 antioxidants-12-01132-f004:**
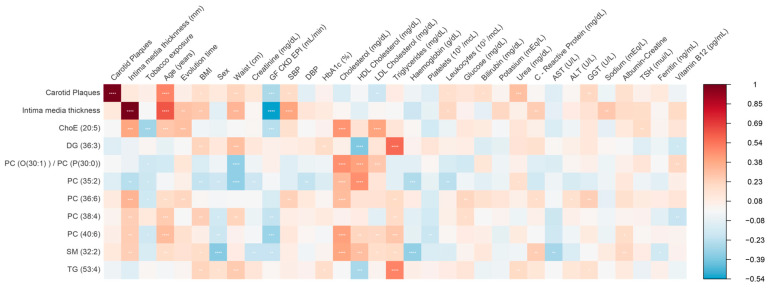
Heatmap of Spearman correlation analyses among lipid species and clinical and analytical parameters. ALT, alanine aminotransferase; AST, aspartate aminotransferase; BMI, body mass index; ChoE, cholesterol ester; DBP; diastolic blood pressure; DG, diglyceride; GGT, gamma-glutamyl transferase; GF CKD EPI, glomerular filtration rate using the CKD-EPI equation; HbA1c, Glycated Hemoglobin; HDL, high-density lipoprotein; LDL, low-density lipoprotein; LPC, lysophosphatidylcholine; PC, phosphatidylcholine; PC(O), alkyl-ether-phosphatidylcholine; PC(P), alkenyl-ether-phosphatidylcholine (plasmalogen); PE, phosphatidylethanolamine; SBP, systolic blood pressure; SM, sphingomyelin; TG, triglyceride; TSH, thyroid-stimulating hormone.

**Table 1 antioxidants-12-01132-t001:** Clinical and biochemical characteristics of study subjects.

	Total (n = 107)	Subclinical Atherosclerosis (n = 52)	Non Subclinical Atherosclerosis (n = 55)	*p*
Age (years)	52.8 (12.9)	60.1 (11.1)	45 (9.81)	<0.001
Diabetes duration (years)	24.5 (11.8)	26.3 (13)	22.6 (10.3)	0.107
BMI (kg/m^2^)	26.7 (4.48)	27.6 (4.49)	25.7 (4.29)	0.027
Central obesity	49.5% (n = 52)	64.2% (n = 34)	34.6% (n = 18)	0.002
HbA1c (%)	7.87 (0.95)	7.87 (0.81)	7.88 (1.10)	0.944
Microvascular complications	54.2% (n = 58)	58.2% (n = 32)	50% (n = 26)	0.396
Hypertension	52.3% (n = 56)	65.5% (n = 36)	38.5% (n = 20)	0.005
Tobacco exposure	41.1% (n = 44)	47.3% (n = 26)	34.6% (n = 18)	0.184
Current smoking	18.7% (n = 20)	21.8% (n = 12)	15.45 (n = 8)	0.394
Tobacco Packs/year	9 (16.4)	11.5 (17.7)	6.37 (14.6)	0.105
Total cholesterol mg/dL	173 (30.9)	170 (31.3)	176 (30.4)	0.291
c-LDL mg/dL	90.2 (24)	86.7 (23.8)	93.9 (24.8)	0.132
c-HDL mg/dL	65.8 (19.2)	65.7 (19.7)	65.9 (18.8)	0.962
Triglycerides	82.6 (43.2)	85.4 (40.3)	79.5 (46.3)	0.483
Glomerular filtrate ml/min	92.3 (17)	86.3 (17.4)	98.6 (14.1)	<0.001
Lipoprotein a	30.9 (29.7)	33.2 (33)	28.4 (25.8)	0.411
CRP	2.26 (3.33)	2.79 (4.11)	1.68 (2.10)	0.105

Data are mean ± standard deviation (continuous variables, *t*-test) or percentage (dichotomous variables, χ^2^ test). BMI, body mass index; CRP, C-reactive protein; c-HDL, high-density lipoprotein cholesterol; c-LDL, low-density lipoprotein cholesterol.

## Data Availability

Not applicable.

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
