# Peer review of "Plasma Lipidomics Profiles Highlight the Associations of the Dual Antioxidant/Pro-oxidant Molecules Sphingomyelin and Phosphatidylcholine with Subclinical Atherosclerosis in Patients with Type 1 Diabetes"

_antioxidants, 2023, doi:10.3390/antiox12051132_

Round 1

Reviewer 1 Report (Previous Reviewer 2)

Yeah you can accept this revision now (the revision is improved and I can understand the limitations for clinical study) 

Yeah you can accept this revision now (the revision is improved and I can understand the limitations for clinical study) 

Author Response

Reviewer 1

Comments and Suggestions for Authors:

Yeah you can accept this revision now (the revision is improved and I can understand the limitations for clinical study)

Response: We thank the reviewer for the positive evaluation of our revision.

Comments on the Quality of English Language:

Yeah you can accept this revision now (the revision is improved and I can understand the limitations for clinical study)

Response: We thank the reviewer for the positive evaluation of our revision.

Reviewer 2 Report (New Reviewer)

Sojo et al. provide an interesting cross-sectional study regarding plasma lipidomics profile of patients with type 1 diabetes.

Before publications, some minor issues should be addressed:

1) The introduction can be further elaborated with inclusion of aspects regarding atherosclerosis, e.g. Yu et al. doi: 10.1042/CS20180911

2) Please describe efforts (statistical/ methodological) to minimize bias. In this context, please list those not addressed as limitation.

3) The microcirculation is known to be altered in patients with diabetes. Please provide subgroup analyses with regard to those patients with microangiopathy and discuss your results. How did you measure microangiopathy? Did you perform for e.g. nailfold microcirculation measurements? Polyneuropathy and nephropathy might not necessarily be caused by microangiopathy. Please mention this as limitation.

3) Please include abbreviations into figure and table legends.

4) The figure legends are too small.

5) There are some passages in bold, which are not titles, please revise.

There are only some minor English language corrections necessary.

Author Response

Reviewer 2

Comments and Suggestions for Authors:

Sojo et al. provide an interesting cross-sectional study regarding plasma lipidomics profile of patients with type 1 diabetes.

Before publications, some minor issues should be addressed:

1) The introduction can be further elaborated with inclusion of aspects regarding atherosclerosis, e.g. Yu et al. doi: 10.1042/CS20180911

Response: Following the reviewer’s suggestions we have now added further elaborated the introduction with reference to the use of lipidomics to study atherosclerosis.

2) Please describe efforts (statistical/ methodological) to minimize bias. In this context, please list those not addressed as limitation.

Response: We have adjusted all our machine learning models for confounding variables known to affect SA and IMT such as age, sex, BMI, diabetes duration, Hba1c, hypertension, glomerular filtrate, smoking, and statin therapy. However, we cannot rule out the contribution of other potential confounders not included in the model. In addition, our study is of cross-sectional nature. Thus, we can only report associations but not infer causality. This is a real world study developed in assistance services. As a result, patients included were not naïve to medications prescribed to control for different cardiovascular risk factors such as statins or antihypertensive treatments. This fact limits the interpretation of surrogate outcomes such as total serum cholesterol and fractions and urine albumin creatinine ratio. We have included all these limitations in a paragraph at the end of the manuscript as limitations of the current study.

3) The microcirculation is known to be altered in patients with diabetes. Please provide subgroup analyses with regard to those patients with microangiopathy and discuss your results. How did you measure microangiopathy? Did you perform for e.g. nailfold microcirculation measurements? Polyneuropathy and nephropathy might not necessarily be caused by microangiopathy. Please mention this as limitation.

Response: As suggested by the reviewer, we have now included subgroup analyses for the lipidomics signatures associated with SA and IMT in patients with and without microvascular complications. The results are shown in the new Figure 3. We have also added a new section 2.4 describing the measurements of microvascular complications. We have substituted “microangiopathy” by “microvascular complications” to avoid any confusion.

4) Please include abbreviations into figure and table legends.

Response: We have now included abbreviations in all figures and the table legends.

5) The figure legends are too small.

Response: We thank the reviewer for this comment. We have now increased the size of all the figure legends.

6) There are some passages in bold, which are not titles, please revise.

Response: We thank again the reviewer for this comment. We have now removed any sentence in bold that is not a title.

Comments on the Quality of English Language:

There are only some minor English language corrections necessary.

Response: We have revised the whole manuscript to correct the English.

This manuscript is a resubmission of an earlier submission. The following is a list of the peer review reports and author responses from that submission.

Round 1

Reviewer 1 Report

·      The authors define sphingomyelin and phosphatidylcholine as antioxidant molecules several times. In my opinion, these molecules cannot be defined so clearly because there are many reports about their controversial role in this respect. For example, stimulation of human coronary smooth muscle cells with SM induces a pro-inflammatory response (doi: 10.1161/ATVBAHA.116.305675)

·      Please enter your Local Ethics Committee approval number

·      What was behind the selection of a specific lipid for analysis (e.g., LPC(18:0); PC(32:0); SM(36:1); DG(36:0); 115 TG(52:3), ChoE(16:0); MAG(18:0). 116 

Why the authors didn't choose for example LPC with which are known as molecules associated with the activation of insulin secretion (e.g. 16:0)?

In this context, the authors should supplement the manuscript with information about the pro- and anti-diabetic effects of the molecules they analyzed. Especially in the case of LPC there i  which is a lot of literature data on the subject ( e.g., doi: 10.1016/j.bbrc.2004.11.120, doi: 10.3390/cells909206210.2147/DMSO.S371370)

·      Please use uniform nomenclature total of 107 with T1DM attending the outpatient clinic of our hospital

·      The quality and resolution of the company is at the moment sufficient please for a significant improvement

·      Correction of several typos is needed, e.g., withT1DM

Author Response

The authors are grateful for the reviewers’ and editor comments which have contributed to clarify the message of our paper and to improve the quality of our submission. The specific comments are addressed below:

The authors define sphingomyelin and phosphatidylcholine as antioxidant molecules several times. In my opinion, these molecules cannot be defined so clearly because there are many reports about their controversial role in this respect. For example, stimulation of human coronary smooth muscle cells with SM induces a pro-inflammatory response (doi: 10.1161/ATVBAHA.116.305675)

Author response:  We fully agree with the reviewer. Many thanks for these appreciations. In fact, we already mentioned the proinflammatory role of sphingomyelins in the discussion. To put further in context these facts, we now also provide this dual role of sphingomyelins in the introduction and discussion.

We have now added this text in the Introduction:

Sphingomyelin has been shown to inhibit the peroxidation of unsaturated phospholipids and cholesterol and was proposed as a natural antioxidant [8, 9]. However, sphingomyelin has also found to show proinflammatory/prooxidant properties (see below in Discussion). On the other hand, phosphatidylcholine also behaves as an antioxidant molecule in atherosclerosis [10], although, again, dual pro / antioxidant actions have also been found, as further developed in the Discussion.

  • Please enter your Local Ethics Committee approval number

Author response: Local Ethics Committee approval number has been included on page 2, line 71.

  • What was behind the selection of a specific lipid for analysis (e.g., LPC(18:0); PC(32:0); SM(36:1); DG(36:0); 115 TG(52:3), ChoE(16:0); MAG(18:0).

Why the authors didn't choose for example LPC with which are known as molecules associated with the activation of insulin secretion (e.g. 16:0)?

In this context, the authors should supplement the manuscript with information about the pro- and anti-diabetic effects of the molecules they analyzed. Especially in the case of LPC there is a lot of literature data on the subject ( e.g.,doi:10.1016/j.bbrc.2004.11.120,doi: 10.3390/cells9092062, 10.2147/DMSO.S371370)

Author response: Thank you for this suggestion. We did not select those lípid species. We put into the model all lipid species and those that emerged as significant in the multivariant, artificial intelligence algorithm (Borutta software) are shown in the figures. On the other hand, the subjects of the study had type 1 diabetes, that by definition  have no residual insulin secretion. This, the hypothesis of lípid species with antidiabetic effects cannot be formulated in this context.

  • Please use uniform nomenclature total of 107 with T1DM attending the outpatient clinic of our hospital

Author response: The reviewer is right and we apologize for this inconsistency. This has been corrected on page 2.

  • The quality and resolution of the company is at the moment sufficient please for a significant improvement

Author response: Thank you for pointing this out. Quality and resolution of images have been improved.

  • Correction of several typos is needed, e.g., withT1DM

Author response: Thank you for pointing this out. Corrections have been made.

Reviewer 2 Report

The authors studied the plasma lipidomics profile of patients with type 1 diabetes (T1DM) and explore the potential associations. They focused on the antioxidant molecules sphingomyelin (SM) and phosphatidylcholine (PC). One hundred and seven patients with T1DM were consecutively recruited. Ultrasound imaging of peripheral arteries was performed using a high image resolution B-mode ultrasound system. Untargeted lipidomics analysis was performed by UHPLC coupled to qTOF/MS. The associations were evaluated using machine-learning algorithms. SM(32:2) and phosphatidylcholine species [PC(30:1) and PC(30:0)] were significantly and positively in subjects with subclinical atherosclerosis (SA). This association was further confirmed in patients with overweight/obesity [specifically with SM(40:2)]. A negative association between SA and lysophosphatidylcholine species was found among lean subjects. In patients without SA, phosphatidylcholines [PC(40:6) and PC(36:6)] and cholesterol esters [ChoE(20:5)] were associated positively with intima media thickness both in subjects with and without overweight/obesity. In summary, the plasma antioxidant molecules SM and PC differed according to the presence of SA and/or overweight status in patients with T1DM. This would be the first study showing the associations in T1DM and the findings may be useful in the targeting of a personalized approach aimed at preventing cardiovascular disease in these patients. While this work is interesting, a number of concerns remain.

1.      The age of the two groups is significantly different, thus creating an aging effect. Blood pressure is another key factor.

2.      This work remains somewhat preliminary with association in nature for the experimental findings. The link between these lipids and AS development should be better interpreted.

3.      Resolution of images is poor. Group labelling cannot be visualized.

4.      Discussion section is written like another introduction.

Author Response

The authors studied the plasma lipidomics profile of patients with type 1 diabetes (T1DM) and explore the potential associations. They focused on the antioxidant molecules sphingomyelin (SM) and phosphatidylcholine (PC). One hundred and seven patients with T1DM were consecutively recruited. Ultrasound imaging of peripheral arteries was performed using a high image resolution B-mode ultrasound system. Untargeted lipidomics analysis was performed by UHPLC coupled to qTOF/MS. The associations were evaluated using machine-learning algorithms. SM(32:2) and phosphatidylcholine species [PC(30:1) and PC(30:0)] were significantly and positively in subjects with subclinical atherosclerosis (SA). This association was further confirmed in patients with overweight/obesity [specifically with SM(40:2)]. A negative association between SA and lysophosphatidylcholine species was found among lean subjects. In patients without SA, phosphatidylcholines [PC(40:6) and PC(36:6)] and cholesterol esters [ChoE(20:5)] were associated positively with intima media thickness both in subjects with and without overweight/obesity. In summary, the plasma antioxidant molecules SM and PC differed according to the presence of SA and/or overweight status in patients with T1DM. This would be the first study showing the associations in T1DM and the findings may be useful in the targeting of a personalized approach aimed at preventing cardiovascular disease in these patients. While this work is interesting, a number of concerns remain.

The authors are grateful for the reviewers’ and editor comments which have contributed to clarify the message of our paper and to improve the quality of our submission. The specific comments are addressed below:

  1. The age of the two groups is significantly different, thus creating an aging effect. Blood pressure is another key factor.

Author response: We fully agree in that both age and blood pressure are important factors to take into consideration. For this reason, we adjusted the analyses for age and presence of hypertension, as previously specified.

  1. This work remains somewhat preliminary with association in nature for the experimental findings. The link between these lipids and AS development should be better interpreted.

Author response: We are sensitive to reviewer comments, reviewing the potential dual proinflammatory/anti-inflammatory role of the different lipid species found and providing a final comment stating that: However, these associations need to be considered preliminary and require further validated in larger cohorts.

Resolution of images is poor. Group labelling cannot be visualized.

 Author response: Thank you for pointing this out. Quality and resolution of images have been improved.

  1. Discussion section is written like another introduction.

 Author response: Many thanks for this comment. It seems that we were not successful in showing the rationale behind the different elements discussed. In fact, there was a hidden structure within the discussion, providing specific comments to each of the main results found. In order to increase the visibility, we have now added different subheadings related to the main parts.

Round 2

Reviewer 1 Report

The authors improved the content of their manuscript.

Author Response

We thank the reviewer.

Reviewer 2 Report

  1. The authors have addressed some of the previous concerns. This work remains preliminary with association in nature for the experimental findings. The link between these lipids and AS development should be better interpreted.

Author Response

We have followed the comments from the reviewer and substantially changed the Discussion. We have also introduced in the title of the manuscript the dual role (pro-antioxidant) of the lipid molecules.

Thank you very much for your attention to our manuscript